# Experimental Study on Self-Centering Performance of the SMA Fiber Reinforced ECC Composite Beam

**DOI:** 10.3390/ma15093062

**Published:** 2022-04-22

**Authors:** Zhao Yang, Tingyu Deng, Jiankun Li, Chengxiang Xu

**Affiliations:** 1School of Urban Construction, Wuhan University of Science and Technology, Wuhan 430065, China; dengty0728@163.com (T.D.); dakun123aa@163.com (J.L.); cx_xu@sina.com (C.X.); 2Institute of High Performance Engineering Structure, Wuhan University of Science and Technology, Wuhan 430065, China

**Keywords:** SMA, ECC, superelastic, self-centering performance, recovery force

## Abstract

The combination of superelastic shape memory alloy fibers and ECC materials can form a new SMA fiber reinforced ECC composite material (SMAF-ECC) with good self-centering performance. In order to study the self-centering performance of the new composite material, 6 groups of pre-notch beam specimens were made for three-point bending cyclic loading tests, and the failure phenomenon, hysteresis curve, self-centering effect and influencing factors of the specimens were analyzed. The research results show that when the SMA fibers are effectively anchored in the ECC matrix, the SMA fibers can exert the superelastic properties to provide the ECC beams with recoverying force, and realize the crack self-closure and deflection self-recovery function for the beams, with the minimum residual crack width and deflection is only 0.9 mm and 1.3 mm respectively. Increasing fiber content can cause a small increase in the self-centering ability of the beams. However, only when the fiber diameter is appropriate, better self-centering effect can be achieved, but the difference caused by fiber diameter in the test was only 5%. SMA Fiber end forms have significant influence on self-centering performance. The knotted end beam can get a more than 70% self-centering ratio, while the straight end beams and bended end beams have no self-centering ability. The research results provide important reference for the research and application of this new self-centering materials and their structures.

## 1. Introduction

In recent years, with the proposing of the concept of seismic resilience, self-centering seismic structures with crack closure ability and small residual deformation have quickly become the research focus of earthquake engineering in the world because they can realize the rapid repair and functional recovery of the structure after earthquake. It is the simplest and effective way to realize the structure self-centering function through high-performance engineering materials.

Engineered cementitious composites (ECC) is a kind of high-performance fiber reinforced cement-based composite with remarkable tensile strain hardening and multiple cracking characteristics. It can provide high ductility, large deformation, high energy dissipation and good durability for engineering structures. Therefore, it is very suitable for seismic structures [1]. However, the high ductility and energy dissipation of ECC is at the cost of significant residual deformation and a large number of crack damage, which is still not conducive to the post-earthquake repair and functional recovery of the structures [2]. Therefore, combining ECC with Shape memory alloy to solve the residual deformation of ECC structure after earthquake has become a new research field of self-centering structure in recent years.

Shape memory alloy (SMA) has unique superelasticity. When applied to seismic structures, the flag shape of hysteretic energy dissipation of SMA superelasticity can effectively improve the energy dissipation capacity of the structure, and its deformation recovery ability after unloading can significantly reduce the residual deformation of the structure after earthquake [3,4,5,6]. However, when SMA is applied directly into concrete structures, the material properties of SMA can not be fully utilized due to the incongruous deformation with concrete and brittle damage of concrete under earthquake. Therefore, using ECC instead of concrete as the matrix material of SMA has become a new research field [7,8,9].

Compared with continuous SMA materials such as bars, rods and strands, SMA fiber (SMAF) has fewer inherent defects and better material properties, no cutting or threading required, easy to be made and lower production cost. Besides, the fiber distribution is uniform, which is more suitable for the wide range cracking of ECC [10]. Some scholars have achieved good self-centering effect by using SMA fiber in cement mortar beams [11,12,13,14]. But due to the inherent brittleness of mortar matrix, the excellent performance of SMAF cannot be fully utilized. However, the combination of high ductility ECC material and superelastic SMAF (SMAF-ECC) can make better use of the excellent performance of both the two materials. On the one hand, SMA fibers can use their own superelastic properties to provide restoring force for ECC materials after earthquakes, such as close cracks and restore deformation. On the other hand, the high ductility of ECC can also achieve better deformation coordination with SMA fibers, and the cracks of ECC are relatively fine, which is also convenient for SMA fibers to close the cracks.

Some scholars have carried out exploratory research on the basic mechanical properties of this new SMAF-ECC composite. Ali et al. [15] studied the mechanical properties of SMAF-ECC material, and the results showed that the tensile and bending properties of SMAF-ECC could be improved significantly, but the compressive strength has not significant improvement, and the cracked SMAF-ECC specimens can self-close after heat treatment. Subsequently, they [10] realized the prestressing effect of SMAF through heat treatment and improved the impact resistance of SMAF-ECC composites. Dehghani et al. [16] compared the mechanical properties of SMAF reinforced self-compacting cement-based composites and steel fiber reinforced composites. The study showed that the mechanical properties of the composites reinforced by SMAF were basically the same as those reinforced by steel fiber. It is also pointed out that a stronger bond between SMAF and cementious matrix is needed to benefit from the unique superelastic properties of SMAF effectively. Subsequently, Dehghani et al. [17] studied the synergistic effect of mixing SMA and PVA fiber in ECC. The study showed that the bending strength and toughness of ECC were improved after mixing SMA and PVA, adding 1.25% SMA could increase the ultimate bending strength by about 51−58%. The compressive strength decreased slightly, but when the fiber content reached 1.25%, the compressive strength began to increase again. Chen et al. [18] studied the crack healing ability of SMA-ECC materials, and found that the SMA-ECC specimens exhibited promising crack healing ability by effectively reducing the crack number and width.

At present, no published data on the self-centering performance of ECC materials strengthened by hyperelastic SMA fiber were found in the literature. Therefore, in this paper, hyperelastic SMA fibers were put in the ECC matrix to fabricate the SMAF-ECC composites and the self-centering performance of this new composite was studied through three-point bending test of pre-notched beams, and the main influencing factors were analyzed as well. The research results have important scientific significance for promoting the development and application of SMAF-ECC, a new type of composite material.

## 2. Experiment Design

### 2.1. Materials

The mixture weight proportion of the ECC specimens used in this study is shown in Table 1. The ECC components include Type I ordinary Portland cement with the measured compressive strength after 28 days of 42.5 MPa, Type F and Class I fly ash according to Chinese standard with the fineness of 43 μm, silica sand with particle sizes ranging between 0.075 mm and 0.15 mm, polycarboxylate superplasticizer (PS), water and polyvinyl alcohol (PVA) fibers. The length of the PVA fiber is 9 mm and the diameter is 31 μm. The tensile strength of the fiber is 1500 MPa, the density is 1.3 g/m^3^, the elastic modulus is 42 GPa, and the maximum elongation ratio is 6%.

Three ECC dog-bone shaped tensile specimens were made according to the Chinese specification JC/T2461-2018: Standard test method for the mechanical properties of ductile fiber reinforced cementitious composites [19], as shown in Figure 1. The ECC material prepared before was poured into the molds to make three tensile specimens. After 24 h, the specimens were removed from the molds and put into the standard curing room for 28 days. The temperature and relative humidity during curing were 20 ± 1 °C and 50 ± 5% respectively.

The uniaxial tensile test was carried out on the tensile specimens by the electro-hydraulic servo universal testing machine. The loading was controlled by displacement with the loading rate of 0.2 mm/min and it was stopped after the main crack appeared in the specimens. The load was recorded by the load sensor of the machine, and the displacement was recorded by the displacement meter arranged on both sides of the specimen. The tensile stress was obtained by calculating the ratio of the load to the cross section area of the specimen, and the tensile strain was obtained by calculating the ratio of the tensile deformation to the gauge length of the specimen.

The tensile stress-strain curve of ECC is shown in Figure 1. It can be seen that the ultimate tensile strain of ECC specimen is more than 3%, and the peak stress is about 3.0 MPa. There are several small cracks appeared in the specimen (Figure 2).

The diameters of the SMA wires used in the test were 0.7, 1.0, and 1.5 mm, respectively. The main material composition of the SMA is 55.86% Ni and 44.14% Ti. In order to ensure the stability of the internal crystal of the SMA wire during the test, the SMA wire was pre-heated, and the SMA wire was placed in boiling water at 100 °C for 15 min, and then taken out to cool. In order to stabilize the mechanical properties of the SMA wire, the heat-treated specimens were placed in boiling water and then ice water for 1 min respectively before the test, and 10 such cooling and heating cycles were performed. Considering that the surface of the SMA fiber is very smooth, it is difficult to form a sufficient strength bonding effect with the ECC matrix. Therefore, in the tests, the SMA fiber end has been made into bended end and knotted end together with the straight end. It is expected that the effective anchorage can be obtained by the end setting.

In order to obtain the mechanical properties of the superelastic SMA wires, the uniaxial direct tensile test and the cyclic tensile test were carried out respectively. The uniaxial tensile stress-strain curve is shown in Figure 3, and the mechanical properties are shown in Table 2, the cyclic tensile stress-strain curve is shown in Figure 4.

### 2.2. Specimen Design and Fabrication

In order to study the self-centering properties of the SMAF-ECC beams, three-point bending test of ECC pre-notch beams under cyclic loading were carried out. The length of the ECC beam specimen is 160 mm, the cross section of the beam is rectangular and the size is 40 mm × 40 mm. In order to ensure that the cracks appearing in middle of the bottom of the ECC beam under bending moment, a PVC plate with 40 mm length, 10 mm width and 1 mm thickness is set at the bottom of the beam formwork, and SMA fibers are placed on the top of the PVC plate. Then the ECC material is poured into the formwork. The SMAF-ECC beam specimens were maintained according to the material test. The specimen design is shown in Figure 5.

In order to study the effects of SMA fiber content, diameter and end form on the self-centering performance of ECC beams, six groups of beam specimens were designed and fabricated. The specific specimen scheme is shown in Table 3, and the form of the SMA fiber end is shown in Figure 6. The total length of the straight SMA fiber is 38 mm, which is placed symmetrically on both sides of the PVC board; the total length of the bended fiber is also 38 mm, the two ends are bent, the bending angle is 45°, and the length of the bending section is 4.0 mm; The distance between two knots of the knotted fiber is 30 mm, and the diameter of the knot is about 8 mm. The comparison specimens E0 were pure ECC specimens without SMAF.

### 2.3. Test Device and Test Method

The loading device diagram of the three-point bending test is shown in Figure 7. The loading point is located in the middle of the top span of the beam specimen, and the axial distance between the two supporting points at the bottom is 120.0 mm. The loading equipment adopts the electro-hydraulic servo universal testing machine, and the vertical load is applied to the midpoint of the beam top through the actuator. The load is measured by the load sensor of the testing machine. The mid-span deflection of the beam specimen is measured by the dial indicator set at the bottom of the beam.

The displacement control was used for cyclic loading, and the loading speed was 0.6 mm/min. The cyclic loading of each stage is (1/10)Δ, where Δ is the mid-span deflection corresponding to the ultimate bearing capacity of the comparison specimen E0. The Δ is 6 mm in the test, and the loading displacement of each stage is 0.6 mm. The specific loading process is shown in Figure 8.

## 3. Test Results

### 3.1. Failure Phenomenon and Analysis

The failure phenomenon in the 10th loading cycle is shown in Figure 9. For the comparison specimen (E0), with the increase of loading deflection, obvious main cracks appeared in the mid-span region of the specimen and the self-centering effect gradually deteriorated after unloading. In the 10th loading cycle, the loading crack width is 3.5 mm, and the residual crack width is still 3.5 mm after unloading. The PVA fibers in the matrix have been pulled out or broken, and they cannot continue to play the role of bridging to achieve the effect of closing cracks. The specimens with straight and bended fiber end (1.0-2-G and 1.0-2-Z) also show poor self-centering performance. At the end of cyclic loading, the loading crack width of 1.0-2-G is 3.6 mm, the residual crack width is 3.6 mm, the loading crack width of 1.0-2-Z is 3.7 mm, and the residual crack width is still 3.7 mm. While the specimens with knotted fiber end (0.7-2-DJ, 1.0-1-DJ, 1.0-2-DJ and 1.5-2-DJ) show good self-centering ability after unloading, the deflection is significantly smaller, the residual crack widths ranges from 0.9 mm to 1.2 mm.

### 3.2. Load-Deflection Curve and Deflection Recovery Curve

Figure 10 is the load-deflection curves of a representative specimen from each group and the corresponding deflection recovery curves. It can be seen that:

The self-centering effect of straight and bended SMAF specimen 1.0-2-G and 1.0-2-Z is not obvious, and there is no flag shape appeared in the load-deflection curve. For specimen 1.0-2-G with bended end SMAF, the main reason is that the stress concentration at the bended ends make the bended ends being stretched straightly, thus the SMA fiber can not get enough anchoring force and it cannot continue to bear more load, thus the stress cannot reach the stress platform level. Therefore, it cannot provide recovery force for the ECC matrix. For specimen 1.0-2-Z with straight end SMAF, the main reason is that the surface of straight SMA fiber is very smooth and cannot provide sufficient anchoring force either, resulting in the continuous debonding of SMA fiber during loading, thus cannot effectively play its superelastic performance. Therefore, the residual deflection of these two kinds of specimens both increase obviously with the loading.

The self-centering effect of the knotted SMAF specimens is good, and there are obvious flag shapes appeared in the load-deflection curves. The main reason is that the knotted SMAF can provide sufficient anchoring force to effectively transfer the recovery force to the matrix. The self-centering effect of the specimen 1.5-2-DJ is the best, and the residual deflection is only 1.3 mm in the 10th cycle with the loading deflection is 6 mm. In the same cycle, the residual deflections of specimen 1.0-2-DJ, 1.0-1-DJ and 0.7-2-DJ are 1.6 mm, 1.8 mm and 2.0 mm respectively.

### 3.3. Influence Factors of Self-Centering Performance

In this study, the effects of SMA fiber content, diameter and end form on the self-centering performance of the beam specimens were analyzed. The self-centering ratio was used to evaluate the self-centering performance. The self-centering ratio u is calculated as Equation (1).
(1)$$u=\frac{\Delta_{1}-\Delta_{0}}{\Delta_{1}},$$
where $\Delta_{1}$ means the maxmum loading deflection in each cycle, $\Delta_{0}$ means the residual deflection after unloading in each cycle. The data for $\Delta_{1}$ and $\Delta_{0}$ comes from the data in Figure 10b.

#### 3.3.1. The Influence of SMA Fiber Content

The self-centering ratio curves of the specimens 1.0-1-DJ and 1.0-2-DJ were selected for comparative analysis (Figure 11a). It can be seen from the figure that the self-centering ratio of the specimen 1.0-2-DJ with two SMA fibers was lower than that of specimen 1.0-1-DJ with one fiber in the first two loading cycles, and both the two self-centering ratio decreased with the increase of load. Because at the beginning of loading, the recovery force mainly comes from the matrix material ECC, and the higher content of SMAF can cause worse influence to the matrix material, resulting in a lower self-centering ratio. With the increase of loading, the superelasticity of SMA fiber begin to provide the recovery force. The larger the fiber content is, the greater the recovery force is. Therefore, the self-centering ratio of specimen 1.0-2-DJ exceeds that of 1.0-1-DJ and the self-centering ratio keeps increasing with the increase of loading. At the last loading cycle, the self-centering ratios of specimen 1.0-2-DJ, begin to decrease and close to that of specimen 1.0-1-DJ, indicating that the anchorage performance of SMA fiber and the matrix in specimen 1.0-2-DJ begins to deteriorate with the increasing deflection. The maxium self-centering ratio of specimen 1.0-2-DJ reaches 75% and the maxium ratio of specimen 1.0-1-DJ reaches 73%. It can be seen from the above analysis that the increase of SMA fiber content can cause a small increase in the self-centering ability of SMAF-ECC composite beam when SMA fiber and matrix have good anchorage and load reaches a certain level, which is close to the results of Lee et al. [20]. But due to the limited experimental data, more tests still need to be done to verify this kind of conclusion.

#### 3.3.2. The Influence of SMA Fiber Diameter

The self-centering ratio curves of specimens 0.7-2-DJ, 1.0-2-DJ and 1.5-2-DJ were selected for comparison. It can be seen from Figure 11b that the three groups of specimens all show good self-centering performance. But the maximum deflection self-centering ratio of specimen 1.0-2-DJ is about 5% higher than that of specimen 0.7-2-DJ and 1.5-2-DJ. This result shows that the SMA fiber diameter is not the bigger the better or the smaller the better. In this test, when the loading displacement is from 1.8 mm to 6 mm, the self-centering rate of the specimen with 1 mm diameter SMA fiber is almost the largest among the three groups of specimens. The above analysis shows that the fiber diameter has a certain influence on the self-centering performance of composite beams. When the fiber diameter is appropriate, better self-centering effect can be achieved.

#### 3.3.3. The Influence of SMA Fiber End Form

The self-centering ratio curves of specimens 1.0-2-Z, 1.0-2-G and 1.0-2-DJ were selected for comparison. It can be seen from Figure 11c that among the three groups of specimens, only the knotted SMA fiber end specimen 1.0-2-DJ showed excellent self-centering effect. The self-centering rate of the bented end specimen 1.0-2-G and the straight end specimen 1.0-2-Z both continue decreasing from the first loading cycle. It is shown that the knotted end can provide effective anchoring force for SMA fiber, so that SMA fiber can exert superelastic characteristics, produce recoverying force and provide self-centering ability for beam specimens. Neither bended ends nor straight ends specimens can provide effective anchorage force for SMAF, so SMA fiber superelasticity cannot be used, and beam specimens cannot obtain self-centering ability. When the loading deflection reaches 6 mm, the self-centering ratio of 1.0-2-DJ is 72.8%, while those ratios of 1.0-2-G and 1.0-2-Z are 2.0% and 2.3% respectively. The above analysis also confirms that knotted fiber end is very effective for improving the anchorage strength of SMA fiber in ECC matrix, stimulating the superelasticity of SMA fiber, and realizing the self-centering function of composite beams. However, neither bended end nor straight end can achieve the above self-centering function, which is consistent with the test results of Choi et al. [21].

## 4. Conclusions

In this paper, the self-centering performance and influencing factors of SMAF-ECC pre-notched beam were studied through three-point bending test. The main conclusions are as follows:When the SMA fibers can be effectively anchored in the ECC matrix, the SMA fibers can fully use its superelastic characteristics and provide flag-shaped hysteretic energy dissipation capability for the SMAF-ECC composite beams. Besides, the recovery force can be also provided by the SMA fibers to the composite beams in the unloading process. Thus makes the composite beams realizing the crack self-closure and deflection self-recovery function. The minimum residual crack width of the composite beam specimens is 0.9 mm, and the minimum residual deflection is 1.3 mm.SMA fiber content, diameter and end form can all affect the self-centering ability of SMAF-ECC composite, but the influence characteristics are different. Increasing fiber content can cause a small increase in the self-centering ability of SMAF-ECC composite beams. However the fiber diameter is not the bigger the better or the smaller the better, only when the fiber diameter is appropriate, better self-centering effect can be achieved. Even so, the self-centering ratio difference caused by different fiber diameters in the experiment was only 5%.SMA Fiber end forms have significant influence on self-centering performance of composite beams. The knotted end beam can get a more than 70% self-centering ratio, while the straight end beams and bended end beams have no self-centering ability.Although the composite beams in this study obtained good self-centering ability, the SMA fiber began to play a role effectively only after the beam had a large deflection. Therefore, how to make SMA fiber play a role earlier is a vproblem to be studied later. In addition, making simpler fiber ends is also needed in subsequent studies.

## Figures and Tables

**Figure 1 materials-15-03062-f001:**
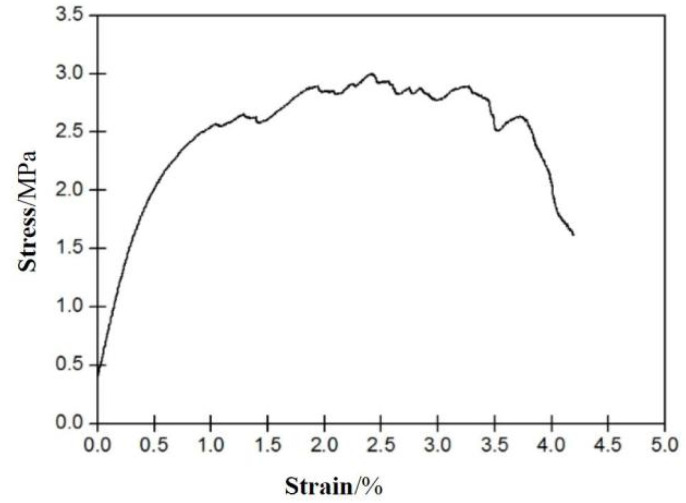
Stress-strain curve of ECC specimen.

**Figure 2 materials-15-03062-f002:**
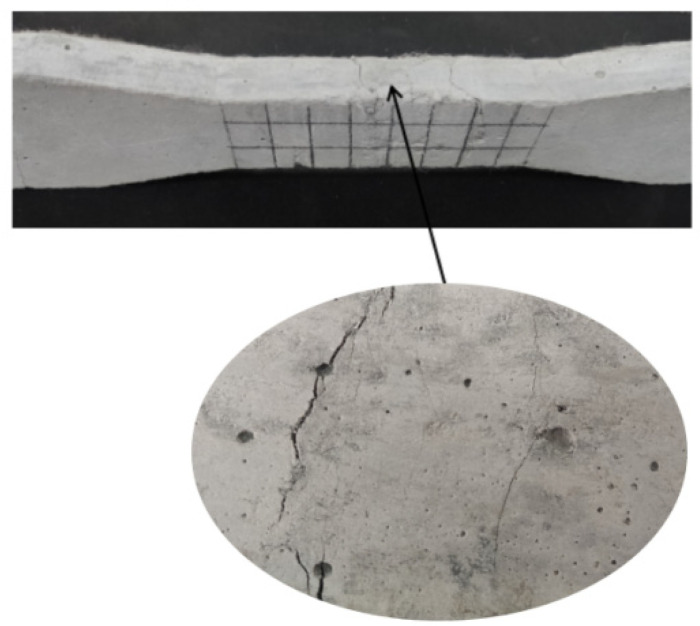
Cracks of ECC specimen.

**Figure 3 materials-15-03062-f003:**
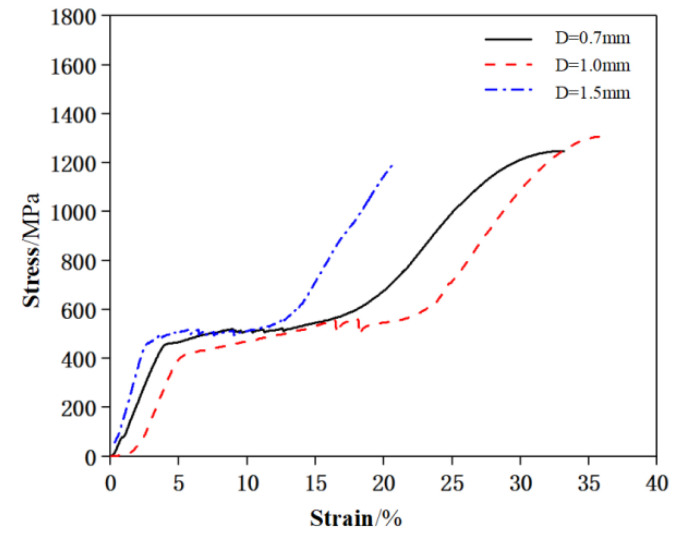
Direct tensile stress-strain curve of SMA wires.

**Figure 4 materials-15-03062-f004:**
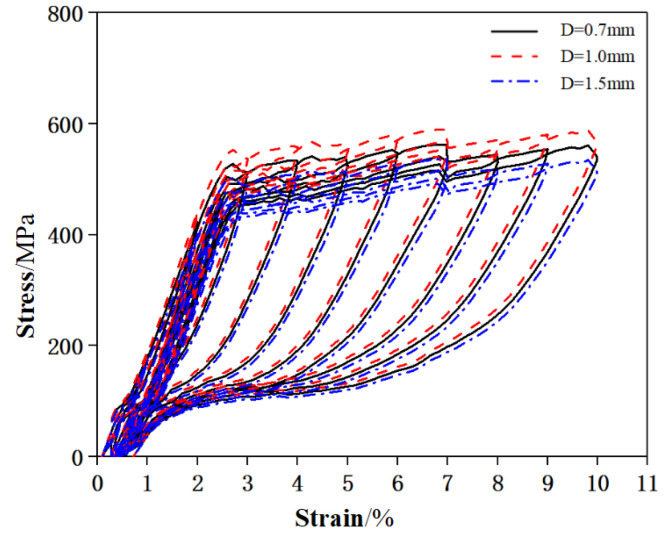
Cyclic tensile stress-strain curve of SMA wires.

**Figure 5 materials-15-03062-f005:**
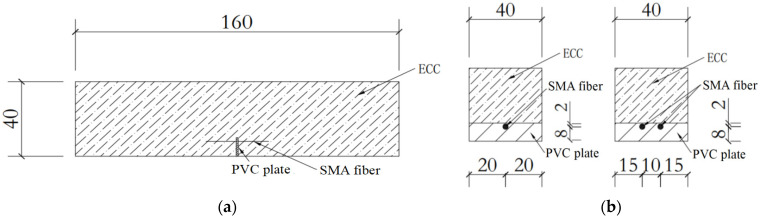
Details of the pre-notch beam specimen/mm: (**a**) Front view of the beam; (**b**) Mid-span section of the beam with sigle and double SMA fiber.

**Figure 6 materials-15-03062-f006:**
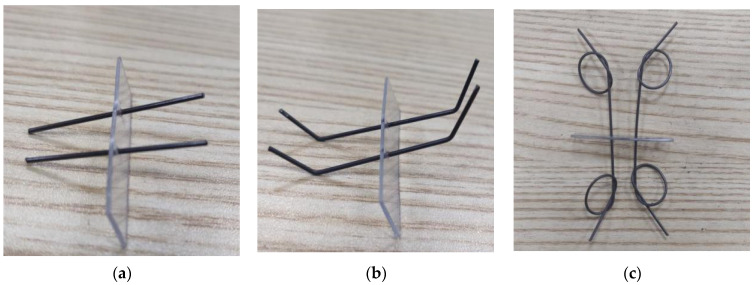
The forms of SMA fiber end: (**a**) straight end; (**b**) bended end; (**c**) knotted end.

**Figure 7 materials-15-03062-f007:**
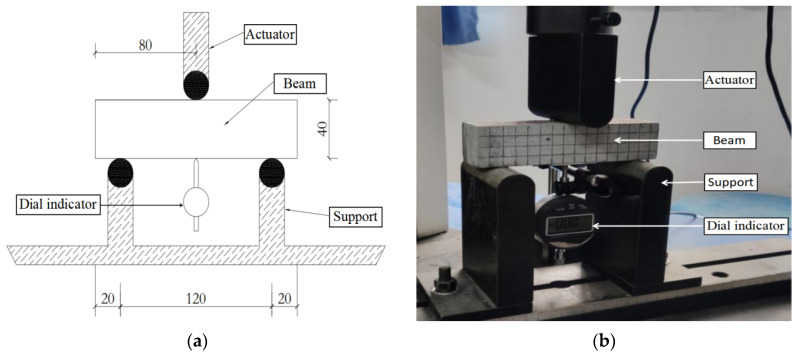
The loading device of the three-point bending test: (**a**) Device diagram; (**b**) Device picture.

**Figure 8 materials-15-03062-f008:**
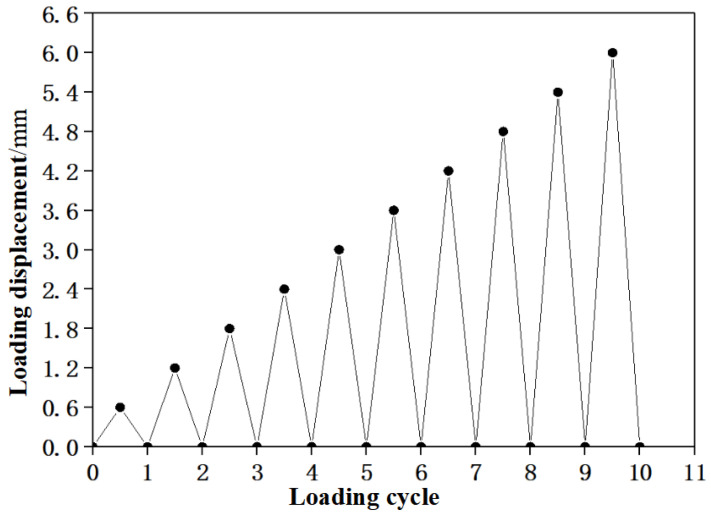
The loading process diagram.

**Figure 9 materials-15-03062-f009:**
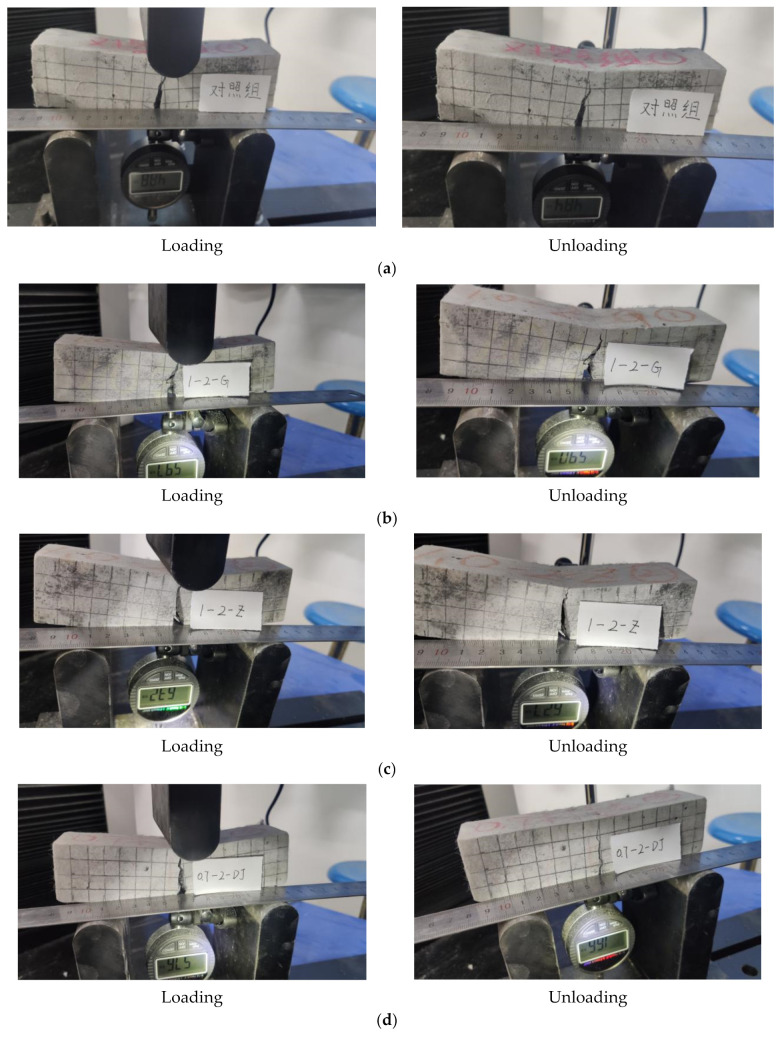
The failure phenomenon in the 10th loading cycle: (**a**) Comparison specimen E0; (**b**) Specimen 1.0-2-G; (**c**) Specimen 1.0-2-Z; (**d**) Specimen 0.7-2-DJ; (**e**) Specimen 1.0-1-DJ; (**f**) Specimen 1.0-2-DJ; (**g**) Specimen 1.5-2-DJ.

**Figure 10 materials-15-03062-f010:**
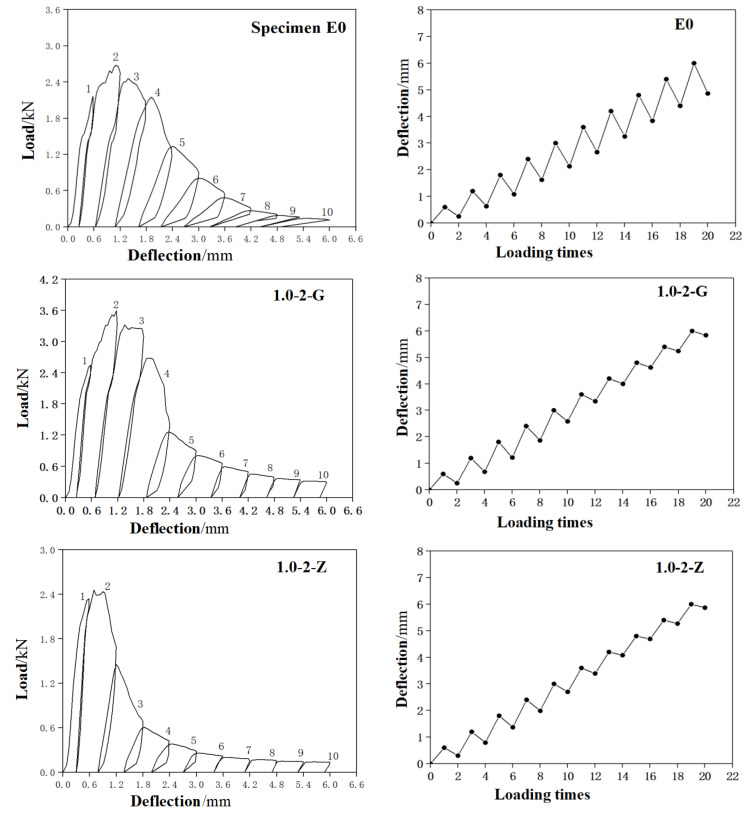
The load-deflection curves and deflection recovery curves: (**a**) load-deflection curves; (**b**) deflection recovery curves.

**Figure 11 materials-15-03062-f011:**
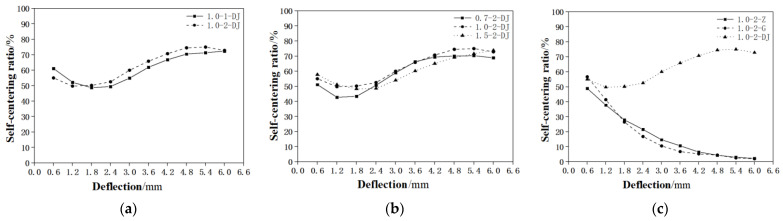
Comparison of the influence on self-centering ratio according to different factors: (**a**) Fiber content; (**b**) Fiber diameter; (**c**) Fiber end form.

**Table 1 materials-15-03062-t001:** Mixture weight proportion of the ECC specimens.

Raw Materials	Cement	Fly Ash	Silica Sand	Water	PS	PVA *(%)
Mix proportion	1.0	2.4	0.36	0.26	0.0082	2.0

* Percentage of fibre content by volume.

**Table 2 materials-15-03062-t002:** Uniaxial tensile mechanical properties of SMA wires.

Diameter/mm	Starting Point of Stress Platform		Ending Point of Stress Platform		Tensile Strength/MPa	Ultimate Strain/%
	Strain/%	Stress/MPa	Strain/%	Stress/MPa		
0.7	4.5	470	18	610	1248	33
1.0	5.2	422	24	610	1307	35
1.5	4.5	450	15	610	1200	21

**Table 3 materials-15-03062-t003:** T Specimen scheme.

Specimen Type	Specimen Name	SMAF Diameter/mm	SMAF Number	SMAF End Form	Specimen Number
	0.7-2-DJ	0.7	2	knotted	3
SMAF reinforced ECC beam	1.0-1-DJ	1	1	knotted	3
	1.0-2-G		2	bended	3
	1.0-2-DJ			knotted	3
	1.0-2-Z			straight	3
	1.5-2-DJ	1.5	2	knotted	3
ECC beam with no SMAF	E0	−	−	−	3

## Data Availability

Not applicable.

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
