# Peer review of "Experimental Study on Self-Centering Performance of the SMA Fiber Reinforced ECC Composite Beam"

_materials, 2022, doi:10.3390/ma15093062_

Round 1
Reviewer 1 Report
The submitted paper deals with the study of composites based on shape memory alloy fibers in cementitious materials. Mechanical properties, failure phenomena, and the self-centering effect are evaluated. The factors influencing the behavior of the studied composite are also discussed.
The paper is well organized, the context of the research is properly elucidated, the results are sufficiently presented and the conclusions are sound.
I would suggest some minor changes before the publication.
- Check the text carefully; there are some typos and unclear sentences that sometimes make the meaning of results and their discussion not easy to understand.
- Differences (and/or similarities) between the results of this study and those reported in the current literature should be highlighted. Besides, the advantages and benefits arising from the investigated materials should be better underlined.
- Lines 115-119 and lines 128-135: since in these parts results are discussed, they should be moved to Section 3.
- Figure 1 and 2 can be merged into one.
Reviewer 2 Report
The paper "Experimental study on self-centering performance of the SMAF-ECC composite beam" is interesting and adheres to the scope of publication of Materials, however, some points must be addressed by the authors before its acceptance:
(1) The abstract needs to include some more detailed quantitative information, this is important for understanding the innovation of this research;
(2) Do not use acronyms in the title, this should be reviewed;
(3) There are numerous figures, they must be rethought, and often grouped together;
(4) Several figures are reduced in size, check this information;
(5) The authors speak of beams, but the specimens are relatively small, looking like prismatic mortars, what would that be?
(6) There are many results, but this is not reflected in the discussions presented, proof of this are the only 18 existing references. Thus, authors should complement the theoretical framework and discussions with other references, such as: 10.1617/s11527-019-1353-x; 10.1016/j.cscm.2021.e00738; 10.1016/j.cscm.2022.e01037.
I suggest approval after minor corrections.
Reviewer 3 Report
This paper focuses on self-centering performance of the SMAF-ECC composite beam. I thinks this is an important result for development an application of SMAF-ECC that needs to be widely circulated. However, several points as indicated below need to be addressed by authors to improve the quality of the articles.
1. p.3 Line120-130
The bond property of SMA wire to ECC has a significant effect on mechanical behavior of the beam specimen. Please add the surface properties of the SMA wire. It is mentioned in the following discussion, but I think the authors should provide the detail information in this section.
2. p.5 Line153-154
“, and the length of the bending section is 4.0mm”.
Is 4.0mm correct? Is it 40mm? Please confirm.
3. pp.9-10 Figure10
Are the load-deflection curves shown as an average of the 3 specimens? If is shows one example of each parameter, please explain it.
4. p.12 Figure11
Is the self-centering ratio the average of 3 specimens? Please add an explanation for this.
And I think the variation needs to be explained. From the results of (a) and (b), it is difficult to determine the effect of fiber content and fiber diameter. It could be considered a range of variation.
5. p.11 Line250-251
“1mm diameter SMA fiber provides the best self-centering performance for the ECC beam.”
Please add a discussion of the reasons for this.
